# OpenReview forum: "An Interpretable N-gram Perplexity Threat Model for Large Language Model Jailbreaks"
_ICML.cc/2025/Conference — ICML 2025 poster_

### Official Review · Reviewer_zGpd · 2025-03-07

**Overall Recommendation:** 1

**Summary:**

This research introduces an interpretable threat model for assessing the vulnerability of LLMs to jailbreaking attacks. The paper proposes using N-gram language model perplexity as a unified, LLM-agnostic metric to evaluate the fluency and likelihood of attacks. It demonstrates that many existing attacks rely on infrequent word combinations. Furthermore, the study shows how this threat model's interpretability allows for a deeper analysis of attack strategies and model weaknesses.

## update after rebuttal
Thank you for the responses and clarification. I am keeping my score mainly because of concerns that may be caused solely by the presentation. I would recommend revising the current version to clarify aspects such as the threat model and evaluation.

**Claims And Evidence:**

I could not identify specific unclear claims, but for more details, check the comments section.

**Essential References Not Discussed:**

Many references are generally published in that domain, but no obvious paper is missing, as far as I can tell.

**Ethical Review Concerns:**

Not necessarily critical, but since the paper is about security, I would expect some comment on that in the paper.

**Ethics Expertise Needed:**

["Privacy and Security"]

**Experimental Designs Or Analyses:**

The soundness is difficult to check as the paper’s goal is not clearly described, making it difficult to assess the experimental setup.

**Methods And Evaluation Criteria:**

The paper shows that the proposed method is reasonable. However, it remains unclear if is new or applicable in varying cases (See comments below).

**Other Comments Or Suggestions:**

Thank you for submitting the paper. It elaborates on a very timely topic with no known solution. Therefore, research in this domain remains required to build more safe and secure LLMs.

**Idea**:

Overall, using perplexity is a good idea. Unfortunately, this is a well-studied countermeasure, and it has also been demonstrated that attacks can be designed to bypass it.

The threat model has the weakness that the attack strategy needs to be known: “threat model checks if a **given** jailbreak is likely to occur in the distribution of text” from the abstract.

The adaptive attack evaluation seems not to be a real adaptive attack but more of a collection of different attacks. For a real adaptive attack, the attacker should know the defense (or parts of it) and can adjust their attack. This helps us study the limitations.

**Presentation:**

Although the general idea is not bad, the presentation makes it difficult to understand the paper's contributions and findings. It is mostly unclear if the paper proposes a defense or an attack.

For example, Figure 1 shows different layers for defense, but the remaining paper does not clarify these layers.

Figure 2 shows unclear results. Specifically, the x-axis is not described. What do FLOPS mean in this context?

The defense is required to choose a threshold for the decision. However, the threshold seems to be fixed. I would expect this threshold to change in cases of different models/attacks or just changing context. Ideally, a threshold is independent of this, or it should be at least evaluated on what effect it has on the results.

Some results are missing in Table 2. Why are no numbers listed for some of the Llama models for BEAST and PAIR?

**Other Strengths And Weaknesses:**

Strengths:

- Timely topic

Weaknesses:

- The scope of the paper remains unclear (attack or defense?)
- The applicability of the approach is not clear

**Questions For Authors:**

- What do FLOPS mean in this context?
- What are the different layers of the defense?
- Why are there no numbers listed for some of the Llama models for BEAST and PAIR?

**Relation To Broader Scientific Literature:**

The paper has a good overview of related work.

**Theoretical Claims:**

N/A

---

> ### Author Rebuttal · Authors · 2025-03-31
>
> Dear reviewer zGpd,
>
> Thank you for your questions and reviewing our paper! We address all of them below.
>
> ------------
>
> **Q: “Is it a defense or an attack paper?”**
>
> **A:** Our paper is neither solely an attack nor a defense paper. We propose a principled framework for comparing attacks by introducing a threat model based on an N-gram perplexity constraint to assess attack fluency in an LLM-agnostic manner and using it to adaptively evaluate attacks.
>
> **Q: “Attack evaluation seems not to be a real adaptive attack … for a real adaptive attack, the attacker should know the defense … ”**
>
> **A:** We agree that adaptive attacks are crucial for fair evaluation. This is exactly what we do: We adapt every attack to the perplexity constraint with full knowledge of the defense (see Section 4 and Appendix F.2). For instance, our adapted PRS attack samples from the top-100k bigrams and applies a filter to every suffix proposal, unlike the original unrestricted token sampling.
>
> This can be more clearly observed by comparing the ASR in the threat model without our adaptive attack (see Figure 12 in Appendix) and with it (see Table 2): E.g., the difference between applying and not applying the adaptive attack in ASR is 82% for PRS.
>
> **Q: “...perplexity is a good idea. Unfortunately, this is well-studied”**
>
> **A:**   We disagree that this countermeasure is well-studied, as every previous perplexity filtering approach lacked the **adaptation of the existing**  attacks (see L152-162).
>
> Our work demonstrates that existing attacks such as PRS and GCG can be adapted to be more effective than attacks incorporating fluency constraint by design (ASR of our adaptive attack for PRS is 82% compared to 19% for AutoDan in Table 2).
>
> **Q: “What are the different layers of the defense?”**
>
> **A:** We agree that this discussion would benefit the main text and we will add it to the final version of the paper:
>
> - *Total FLOPs*: (**fl**oating point **op**eration**s**) represent a hardware-agnostic measure of computational budget from the attacker’s perspective. It captures the total computational effort needed to create a jailbreak, — making some attacks prohibitively expensive and thus lowering their ASR (please see detailed discussion in Appendix E).
> - *N-gram Perplexity*: constraint ensures that the attacker maintains input fluency, further reducing ASR (Figure 5).
> - *System Prompt*: constraint ensures that all attacks are evaluated under a “safe system prompt,” which, as mentioned in Related Work (L132-136), serves as an effective mitigation strategy.
>
> To further clarify the importance of these components, we ablate them for PRS and Llama2-7B in the following table:
>
> | ASR  | System Prompt | FLOPs < 5x10¹⁵ | N-PPL < γ₀.₉₉₉ |
> |--------|-------------|----------------|------------------|
> | 0.98   | ✗           | ✗              | ✗                |
> | 0.90   | ✓           | ✗              | ✗                |
> | 0.80   | ✓           | ✓              | ✗                |
> | 0.50   | ✓           | ✓              | ✓                |
>
> **Q: “Why are there no numbers listed for some of the Llama models for BEAST and PAIR?”**
>
> **A:** In our study, we first identified GCG, PRS, and AutoDan as the best-performing attacks (see Figure 2). Due to computational limitations in our academic lab, we focused our evaluation on these attacks for the more recent safety-tuned models during our submission.
> After the submission, we finished the evaluation for the rest of the models for BEAST and PAIR:
> | LLM (Elo ↑)      | BEAST | + A | PAIR | + A |
> |----------------------|----------:|--------:|---------:|--------:|
> | Llama3-8b (1152)     | 0.02      | 0.01    | 0.02     | 0.03    |
> | Llama3.1-8b (1171)   | 0.05      | 0.06    | 0.04     | 0.03    |
> | Llama3.2-1b (1061)   | 0.14      | 0.14    | 0.02     | 0.05    |
> | Llama3.2-3b (1105)   | 0.14      | 0.14    | 0.15     | 0.15    |
> | Gemma2-2b (1136)     | 0.10      | 0.10    | 0.27     | 0.27    |
> | Starling-7b-α (1088) | 0.16      | 0.15    | 0.51     | 0.51    |
> | **Overall Average ASR**      | 0.10  | 0.10 | 0.13 | 0.13|
>
> These additional results confirm the original trend, with PAIR and BEAST significantly underperforming compared to adapted versions of PRS and GCG.
>
> **Q: “...threat model has the weakness that the attack strategy needs to be known”**
>
> **A:** The perplexity filter together with its threshold are chosen independently of any specific attack strategy (see Section 3.2).
>
> **Q: “Ethical review concerns”**
>
> **A:** We address ethical considerations in the Impact Statement on page 9. Please, let us know if you think that some important part of the discussion is missing over there.
>
> -------
>
> Overall, we thank you for your detailed feedback and for highlighting areas where our presentation could be clearer. We are happy to address any further questions and kindly ask you to consider raising your score if our clarifications meet your expectations.

---

> > ### Comment · Reviewer_zGpd · 2025-04-05
> >
> > Dear authors,
> >
> > Thank you for the response. I can understand the explanation, but I still find it hard to change my opinion.
> >
> > Specifically:
> >
> > - It is okay to investigate both attacks and defenses. But it should be clear from the paper what the goal is and if this is changing
> > - I would argue the proposed adaptive adapt is not really adaptive but just a *stronger* attack. For an adaptive attack, an attacker would make use of the changes in parameters influenced by the defense in order to invert it. In this case, we are just sampling from a set that could be received from any source.
> > - What terms like FLOP are is general knowledge. However, it is unclear why it is important here and used to measure the effectiveness of the attack.

---

> > > ### Author Response · Authors · 2025-04-05
> > >
> > > Dear Reviewer,
> > >
> > > **We propose a threat model (which can be viewed as a defense).** We are sorry that there is confusion about whether this is an attack or defense paper, but unfortunately, we cannot track the origin of this misunderstanding. We propose a threat model (which can be viewed as a defense). Then, we benchmark many popular attacks under this threat model. We adapt the attacks to the threat model — this is a standard practice when evaluating attacks under new constraints.
> > >
> > > **We do adaptive attacks with full knowledge of the threat model.** For example, vanilla PRS (without N-gram perplexity filter) achieves 89% ASR but drops to 0% ASR when using the N-gram perplexity filter. Our PRS, adapted to the N-gram perplexity filter (in short, adaptive), still achieves an ASR of 82%, **not because it is a stronger attack** but because it has been adapted with full knowledge of the threat model (the N-gram perplexity filter). That is, we are not sampling from *any set*, but i) we sample from the *exact set* of most frequent bigrams used in the threat model, and ii) we select candidates only those passing the filter.
> > >
> > > **Increasing FLOPs increases ASR.** Every attack improves with more compute, so comparisons must be made under equal budgets. **FLOPs do not measure effectiveness**. FLOPs measure the compute required to reach a certain effectiveness — ASR. This makes FLOPs a fair, hardware-independent way to assess how hard it is to run a certain attack — not how good it is. We encourage you to see our discussion with Reviewer FYyk and Appendix E for further details on why we use FLOPs as a measure of compute budget.

---

### Official Review · Reviewer_6bX4 · 2025-03-12

**Overall Recommendation:** 4

**Summary:**

This paper presents a fundamental formulation of jailbreaking attacks, i.e. a unified threat model for them. Leveraging the N-gram language model theory, the proposed technique successfully constructs a threat model and demonstrates a successful defense against multiple attacks.

## update after rebuttal

My concerns are addressed and still leaning toward acceptance.

**Claims And Evidence:**

The N-gram model is well-formulated and presents effectiveness with evaluations under adaptive attacks.

**Essential References Not Discussed:**

N/A

**Experimental Designs Or Analyses:**

The evaluation is comprehensive, covering multiple jailbreaking attacks. However, I’m also curious about how the proposed PPL evaluated under other attacks with natural languages like in-context learning-based [1].

[1] Jailbreak and Guard Aligned Language Models with Only Few In-Context Demonstrations  https://arxiv.org/pdf/2310.06387

**Methods And Evaluation Criteria:**

Yes.

**Other Comments Or Suggestions:**

N/A

**Other Strengths And Weaknesses:**

The focused problem is a very important contribution to the safety community, and the evaluation is comprehensive.

**Questions For Authors:**

See above

**Relation To Broader Scientific Literature:**

This paper may demonstrate broader understanding regarding LLM safety.

**Theoretical Claims:**

Not applied.

---

> ### Author Rebuttal · Authors · 2025-03-31
>
> Dear reviewer 6bX4,
>
> ------------
>
> Thank you for your review and for the high assessment of our work. We would be happy to answer your question regarding the In-Context Attacks (ICA).
>
> The best-performing attack, PRS [1a], builds upon and cites [1]. As [1], it relies on an in-context template that includes an outline of the valid answer. We found that certain parts of the original template trigger the perplexity filter, necessitating their removal or adaptation.
>
> Because the attacker is in full control over added demonstrations, they can select those which do not trigger the filter. Therefore, an ICA attack remains a valid strategy under our proposed threat model, and we expect its attack success rate (ASR) to remain largely unaffected.
>
> We hope that this answers your question!
>
> **References**
>
> [1a]  Andriushchenko et al, Jailbreaking Leading Safety-Aligned LLMs with Simple Adaptive Attacks, ICLR 2025.

---

### Official Review · Reviewer_FYyk · 2025-03-12

**Overall Recommendation:** 2

**Summary:**

This paper proposes an interpretable threat model for evaluating jailbreak attacks on large language models by leveraging N-gram perplexity as a measure of text fluency. By constructing a lightweight N-gram language model on a trillion-token subset of the Dolma dataset, the approach enables LLM-agnostic and computationally efficient evaluation of adversarial inputs, providing clear interpretability by analyzing individual N-gram contributions. The authors adapt several existing jailbreak attack methods—such as PRS and GCG—to operate within this model, and their experiments demonstrate that even when constrained by perplexity filters, discrete optimization-based attacks can maintain high success rates. Additionally, the paper highlights that N-gram perplexity offers significant advantages over LLM-based self-perplexity in terms of cross-model comparability and efficiency, ultimately offering a robust framework for assessing and enhancing the safety of LLMs.

**Claims And Evidence:**

1. "rigorous comparison": The abstract claims that the comparison is rigorous, but how is a rigorous comparison defined? What kind of comparison can be identified as rigorous?

2. Unproven Upper-Bound Claim: The claim that N-gram perplexity effectively upper bounds LLM-based self-perplexity is supported primarily by experimental observations rather than a formal theoretical derivation or proof, leaving the underlying conditions for this relationship unclarified.

3. Interpretability: While the paper asserts that the threat model is inherently interpretable—allowing analysis of individual N-gram contributions—it lacks a formal treatment or proof that quantitatively links these contributions to the rarity in the training data and to the effectiveness of the filter.

**Essential References Not Discussed:**

Please see my aforementioned related papers.

**Experimental Designs Or Analyses:**

1. Limited Dataset and Model Selection: The evaluation primarily relies on 300 malicious queries from a single dataset (HarmBench) and a limited set of models, mainly from the LLaMA and Gemma families. This may restrict the generalizability of the findings to other datasets or LLM architectures. To enhance robustness, consider conducting experiments on additional datasets such as [1-2] and evaluating more models, including OpenAI’s o1, o3, and DeepSeek-R1.

2. Comparison with Multi-Turn Jailbreak Methods: Given the rise of multi-turn jailbreak methods [3-5], it is important to clarify whether these attacks fall within the proposed threat model.

3. Evaluation Bias in Automated Judging: The jailbreak success rate is assessed primarily using an automated judge, a fine-tuned LLaMA2-13B model. However, this approach may not fully capture the nuances of harmful content as perceived by human evaluators. To ensure robustness and alignment with real-world perceptions, consider complementing the automated evaluation with extensive human assessments, particularly for borderline cases.

4. Computational Cost Measurement: Using FLOPs as a proxy for computational cost may oversimplify the evaluation, as FLOPs do not always translate directly to practical runtime or efficiency across diverse hardware setups. To provide a more comprehensive assessment, consider supplementing FLOP-based metrics with actual runtime measurements on different hardware configurations. This would offer a more accurate representation of the real-world computational cost.

[1]. Shen X, Chen Z, Backes M, et al. " do anything now": Characterizing and evaluating in-the-wild jailbreak prompts on large language models[C]//Proceedings of the 2024 on ACM SIGSAC Conference on Computer and Communications Security. 2024: 1671-1685.

[2]. Jin H, Zhou A, Menke J, et al. Jailbreaking large language models against moderation guardrails via cipher characters[J]. Advances in Neural Information Processing Systems, 2024, 37: 59408-59435.

[3]. Ren Q, Li H, Liu D, et al. Derail Yourself: Multi-turn LLM Jailbreak Attack through Self-discovered Clues[J]. arXiv preprint arXiv:2410.10700, 2024.

[4]. Sun X, Zhang D, Yang D, et al. Multi-Turn Context Jailbreak Attack on Large Language Models From First Principles[J]. arXiv preprint arXiv:2408.04686, 2024.

[5]. Russinovich M, Salem A, Eldan R. Great, now write an article about that: The crescendo multi-turn llm jailbreak attack[J]. arXiv preprint arXiv:2404.01833, 2024.

**Methods And Evaluation Criteria:**

The overall methodology and evaluation are reasonable and well-balanced.

**Other Comments Or Suggestions:**

No.

**Other Strengths And Weaknesses:**

Well-structured paper with clear and compelling writing.

**Questions For Authors:**

Please see my aforementioned comments.

**Relation To Broader Scientific Literature:**

The proposed methods and evaluation criteria appear well-aligned with the goal of assessing LLM jailbreak attacks. The use of an interpretable N-gram perplexity threat model provides a clear, LLM-agnostic metric for text fluency, which is essential for comparing attacks across different models. Additionally, benchmarking against established datasets like Dolma and using metrics such as attack success rate (ASR) and computational cost (measured in FLOPs) offer a comprehensive evaluation framework.

**Theoretical Claims:**

The paper defines a jailbreak and the threat model using N-gram perplexity in a mathematically intuitive way (e.g., Equations (2) and (3)), but it does not provide formal proofs to demonstrate that these definitions rigorously capture the intended security properties.

---

> ### Author Rebuttal · Authors · 2025-03-31
>
> Dear reviewer FYyk,
>
> Thank you for your review and the interest in our work. Below, we answer your questions.
>
> -----------
>
> **Q: “Supplementary Material”**
>
> **A:** Our code is available here: https://anonymous.4open.science/r/llm-threat-model-57C3/README.md
>
> Furthermore, we believe that there might be some confusion, as we accompanied our paper with Appendix, which starts on page 12 and contains extended experimental details such as Human Evaluation (Appendix B), Transfer to API models (Appendix H), discussion of FLOPs (Appendix E) and further details regarding the adaptive attacks and N-LM.
>
> Please, let us know, if you meant something else with extended experiment details.
>
> **Q: “Why rigorous / Upper-Bound / Interpretability”**
>
> **A:** We think our work provides a more thoughtful and fair comparison than previous efforts. For instance, we investigate the effects of window and N-gram sizes in App. C, provide adapted versions of existing jailbreaking attacks against the N-gram perplexity filter in App. F, and ablate the effect of tightening the threat model in Section 5.4. We are sorry that "rigorous" caused some misinterpretation and we will get rid of it in the final version. Additionally, we will change the caption of Figure 6 to clarify that we refer solely to the empirical upper-bounding of LLM perplexity by N-gram perplexity. Similarly, our claim regarding interpretability is based on empirical observations: our threat model tracks the influence of every bigram on the resulting perplexity, and we observe that restricting attacks from using infrequent bigrams results in higher computational effort and lower ASR.
>
>
> **Q: “Using FLOPs as a proxy for computational cost may oversimplify the evaluation”**
>
> **A:**  While runtime  measurements are indeed more reflective of real-world performance efficiency, they, unlike FLOPs, are highly dependent on hardware configuration and implementation specifics.
>
> To illustrate this limitation of walltime, we conducted a small ablation on available GPUs using the vanilla GCG attack on Llama2-7B. Table 1 below compares the time per GCG step and corresponding FLOPs across four different GPUs:
>
>
> | GPU (NVIDIA)                    | Time per GCG step (s)    | FLOPs per GCG step                           |
> |------------------------|--------------------------|----------------------------------------------|
> | A100 40GB       | 9.29 ± 0.18              | 1.47×10¹⁵ ± 3.69×10¹³                        |
> | L40S 48GB       | 11.15 ± 0.20             | 1.47×10¹⁵ ± 3.16×10¹³                        |
> | RTX 6000 50GB   | 13.09 ± 0.42             | 1.44×10¹⁵ ± 4.06×10¹³                      |
> | L4 24GB (2)     | 33.62 ± 0.79             | 1.44×10¹⁵ ± 2.67×10¹³                        |
>
> While the FLOP count remains nearly consistent across GPUs, the actual runtime can vary substantially. Therefore, for a fair comparison across different hardware setups and due to the compute constraint, we chose to rely on FLOPs.
>
> **Q: “Dataset and model choice”**
>
> **A:** We appreciate the reviewer’s comment and agree that adding datasets could enhance our evaluation's robustness. However, jailbreaking attacks on LLMs are much more computationally intensive than standard adversarial robustness experiments—a single method–model–behavior combination requires several GPU-hours, and our study already pushes the limits of what a small academic lab can manage with 300 behaviors per cell in Table 1, with more than 30k GPU-hours spent in total.
>
> Moreover, the multi-turn methods [3] and [5] evaluate on only 50 (HarmBench) and 150 (HarmBench and AdvBench) behaviors, respectively. We chose HarmBench because it spans both contextual and non-contextual behaviors across diverse categories.
>
> Finally, we evaluated attacks under this threat model on a variety of large closed- and open-source or large models - GPT-3.5, GPT-4o, GPT-4 Turbo, Llama-3.1-405B, Hermes-3 Llama-3.1-405B, and WizardLM-2-8x22B - which we included in Appendix H.
>
> **Q: “What about multi-turn methods?”**
>
> **A:** To the best of our knowledge, multi-turn methods, including [3-5] are, in spirit, similar to PAIR — an approach largely unaffected by our N-LM filter since it is designed to produce fluent prompts. Therefore, we consider multi-turn attacks to be a valid strategy within our threat model, where at each turn a new attacker's query is evaluated as if it were a single-turn attack, with rejection occurring on a per-query basis.
>
> **Q: “Bias in LM Judging”**
>
> **A:** We share your concerns regarding LM-based judging. Therefore, we rely on the HarmBench judge, which shows high human agreement rates and established benchmarking. In Appendix B, a human study with labeling 2k responses confirmed that HarmBench judge is the most accurate among those that we evaluated.
>
>  -----
>
> Thank you again for your constructive feedback. If you think that we have addressed some of the points you raised, we would kindly ask you to consider raising your score.

---

### Official Review · Reviewer_X1bx · 2025-03-14

**Overall Recommendation:** 3

**Summary:**

This paper introduces an interpretable threat model for evaluating jailbreak attacks on Large Language Models (LLMs), using N-gram language model perplexity as a unified fluency metric. Specificailly, a lightweight LLM-agnostic bigram model has been built for providing interpretability, computational efficiency, and transparency. Popular jailbreak attacks (GCG, PRS, AutoDan, BEAST, PAIR) were adapted and benchmarked across safety-tuned LLMs. Results show that discrete optimization attacks (PRS, GCG) significantly outperform LLM-based attacks, even under fluency constraints. The model's interpretability reveals that successful attacks often leverage infrequent words or domain-specific language (e.g., Reddit, code). Crucially, the paper demonstrates that relying on computational burden (self-perplexity) for security can be misleading.

**Claims And Evidence:**

Overall, most claims have been supported by convincing evidence. However, the claim that self-perplexity defenses provide "security by obscurity" rather than true robustness has reasonable evidence. Table 3 shows adaptive attacks against self-perplexity achieving similar ASR as those against N-gram perplexity but with significantly higher computational costs. The inference space appears more constrained than complete security, though the distributional analysis could benefit from additional experimental validation.

**Essential References Not Discussed:**

Nil.

**Experimental Designs Or Analyses:**

- One issue with the experimental design is the apparent contradiction between the statement in Section 5.2 that Gemma-7b is the most robust model and the ASR numbers in Table 2 where Llama2-7b appears to have lower ASR for several attacks. This inconsistency deserves clarification.

- The human evaluation described in Appendix B provides important validation that the selected judge (Llama2-13B) has a high human agreement (92% accuracy), but this critical validation should have been mentioned in the main paper, given its importance to the overall results.

**Methods And Evaluation Criteria:**

Yes. But one potential limitation of the evaluations is that they focus primarily on English-language models and threats. This work could be strengthened by investigating whether the N-gram approach generalizes across languages, particularly those with different morphological structures.

**Other Comments Or Suggestions:**

- The human evaluation results described in Appendix B are suggested to be integrated into the main text to emphasize the reliability of the judge model used for assessing jailbreak success.
- Expanding the analysis to non-English datasets would enhance the paper's applicability across diverse linguistic contexts.

**Other Strengths And Weaknesses:**

### Strengths:
- The paper is well written.
- The proposed approach removes restrictive assumptions about neural network-based perplexity, making it both interpretable and computationally efficient.

### Weaknesses:
- The evaluation focuses primarily on English-language models, leaving questions about applicability to other languages or domains with different linguistic structures.
- While Figure 13 highlights the successful transfer of jailbreaks to other models, it does not sufficiently analyze why certain models (e.g., Meta-Llama3.1-405b-Instruct) exhibit lower transfer ASR despite extensive safety fine-tuning. Understanding these nuances could inform better defensive strategies against adaptive attacks

**Questions For Authors:**

- How sensitive is the TPR or N-gram LM perplexity to variations in training dataset composition? Would different datasets significantly alter its effectiveness?
- Can the proposed threat model generalize to languages with rich morphology or free word order? If not, what modifications would be necessary?
- Table 2 suggests that Gemma-7b is less robust than some Llama models despite being described as highly robust in Section 5.2. Could you clarify this inconsistency?

**Relation To Broader Scientific Literature:**

The proposed jailbreaking benchmarking methodology complements efforts like HarmBench, which standardizes attack evaluations but lacks robust adaptive attack comparisons.

**Theoretical Claims:**

Not applicable.

---

> ### Author Rebuttal · Authors · 2025-03-31
>
> Dear Reviewer X1bx,
>
> Thank you for your review and the positive assessment of our work. We would like to address the questions you raised.
>
> -------
>
> **Q: “Table 2 suggests that Gemma-7b is less robust than some Llama models…”**
>
> **A:** Thank you for pointing this out, we are happy to provide clarification. What we meant is that Gemma-7b is the most robust model under the strongest attack (PRS). It shows an ASR of 0.45 in its vanilla version and 0.46 in its adaptive version—by far the lowest success rate among the evaluated models for PRS. We will adapt the main text to make it more clear.
>
> **Q: “Main text would benefit from human evaluation results”**
>
> **A:** Thank you for the suggestion: We put the human evaluation results (including the 92% judge agreement rate) from the current Appendix B into the main text in the final version of our paper to better highlight the reliability of our evaluation.
>
> **Q: “How sensitive is the TPR or N-gram LM perplexity to variations in training dataset composition? Would different datasets significantly alter its effectiveness?”**
>
> **A:** *TPR sensitivity:* Our N-gram language model is built on the Dolma ("training set"), and the exact perplexity threshold is selected on AlpacaEval ("validation set") to ensure high utility in chat scenarios. Changes in the training distribution affect the exact PPL threshold, but the TPR remains fixed. Figure 5 shows how different TPR and thresholds impact resulting ASR.
>
> *NLM perplexity sensitivity:* Figure 7a in the Appendix shows that rejection rates across Dolma and AlpacaEval datasets are stable for a fixed NLM perplexity, which hints that perplexity quantiles used in our method are not sensitive to dataset composition, ensuring effectiveness.
>
>
> **Q: “Can the proposed threat model generalize to languages with rich morphology or free word order? If not, what modifications would be necessary?”**
>
> **A:** You raise an excellent point. The focus on English is a common limitation in many jailbreaking studies and benchmarks. This underexplored area has allowed some attacks to exploit poor safety generalisation in rare languages [1], which complicates direct ASR comparison for different languages, and even mere translating HarmBench queries into other languages can itself be seen as an “attack”.
> To assess our constructed N-gram filter utility across languages, we translated 300 HarmBench queries into several target languages with varying morphologies and observed the following rejection rates:
>
> | Morphologically Hard        | Percentage (%) | Morphologically Simple | Percentage (%) |
> |-----------------------------|----------------|------------------------|----------------|
> | Finnish                     | 68.7%          | German                 | 29.0%          |
> | Hungarian                   | 60.3%          | Spanish                | 26.3%          |
> | Czech                       | 51.7%          | French                 | 37.7%          |
> | Polish                      | 52.3%          |  Japanese               | 0.3%           |
> | Turkish                     | 17.3%          | Korean                 | 0.0%         |
> |Russian                | 1.0%                 | Chinese                | 0.0%          |
> | **Average (Hard)**          | **41.8%**      | **Average (Simple)**   | **15.5%**      |
>
> We observe that our N-gram filter generalizes surprisingly well to a variety of languages (on average indeed worse to morphologically richer ones) despite being based on Dolma - officially an English-only dataset. This of course means that some other languages were included, but not filtered in the dataset.
> For future jailbreaking benchmarks covering a diverse range of languages, ensuring balanced language representation in the training dataset will be essential to preserve the filter’s overall utility and effectiveness. More crucially, it is necessary to employ tokenizers that account for the unique features of each language, as current English-centric tokenizers have been shown to severely affect language modeling performance [2]. Both of these we view as orthogonal research directions.
>
> **References:**
>
> [1] Deng, et al. (2024). Multilingual jailbreak challenges in large language models. arXiv. https://arxiv.org/abs/2310.06474
>
> [2] Arnett, et al. (2024). Why do language models perform worse for morphologically complex languages? arXiv. https://arxiv.org/abs/2411.14198
>
> -----
> Thank you again for your constructive feedback! We are happy to answer any further questions. If you think that our responses have addressed your concerns, we would appreciate it if you could consider increasing your score.

---

### Decision · Program_Chairs · 2025-05-01

**Decision:**

Accept (poster)

**Comment:**

This paper introduces an interpretable threat model for evaluating jailbreak attacks on large language models (LLMs). By leveraging N-gram language perplexity, the authors achieve computational efficiency and enhance transparency in their evaluation, contributing significantly to the understanding of jailbreak attacks and the development of effective defenses. However, the paper would benefit from further clarification and a more in-depth analysis of the interpretability aspect.